# Longitudinal Imaging Using PET/CT with Collagen-I PET-Tracer and MRI for Assessment of Fibrotic and Inflammatory Lesions in a Rat Lung Injury Model

**DOI:** 10.3390/jcm9113706

**Published:** 2020-11-18

**Authors:** Irma Mahmutovic Persson, Nina Fransén Pettersson, Jian Liu, Hanna Falk Håkansson, Anders Örbom, René In ’t Zandt, Ritha Gidlöf, Marie Sydoff, Karin von Wachenfeldt, Lars E. Olsson

**Affiliations:** 1Department of Medical Radiation Physics, Institution of Translational Medicine, Faculty of Medicine, Lund University, 20502 Malmö, Sweden; lars_e.olsson@med.lu.se; 2Truly Labs, Medicon Village, 223 63 Lund, Sweden; nina@trulylabs.com (N.F.P.); Jian@trulylabs.com (J.L.); hanna@trulylabs.com (H.F.H.); karin@trulylabs.com (K.v.W.); 3Department of Oncology and Pathology, Clinical Sciences, Lund University, 22184 84 Lund, Sweden; anders.orbom@med.lu.se; 4Lund University BioImaging Centre, Faculty of Medicine, Lund University, 221 42 Lund, Sweden; rene.in_t_zandt@med.lu.se (R.I.Z.); ritha.gidlof@med.lu.se (R.G.); marie.sydoff@med.lu.se (M.S.)

**Keywords:** lung imaging, interstitial lung disease (ILD), drug toxicity, Collagen-I, fibrosis, drug-induced interstitial lung disease (DIILD), magnetic resonance imaging (MRI), positron emission tomography (PET), animal model, bleomycin

## Abstract

Non-invasive imaging biomarkers (IBs) are warranted to enable improved diagnostics and follow-up monitoring of interstitial lung disease (ILD) including drug-induced ILD (DIILD). Of special interest are IB, which can characterize and differentiate acute inflammation from fibrosis. The aim of the present study was to evaluate a PET-tracer specific for Collagen-I, combined with multi-echo MRI, in a rat model of DIILD. Rats were challenged intratracheally with bleomycin, and subsequently followed by MRI and PET/CT for four weeks. PET imaging demonstrated a significantly increased uptake of the collagen tracer in the lungs of challenged rats compared to controls. This was confirmed by MRI characterization of the lesions as edema or fibrotic tissue. The uptake of tracer did not show complete spatial overlap with the lesions identified by MRI. Instead, the tracer signal appeared at the borderline between lesion and healthy tissue. Histological tissue staining, fibrosis scoring, lysyl oxidase activity measurements, and gene expression markers all confirmed establishing fibrosis over time. In conclusion, the novel PET tracer for Collagen-I combined with multi-echo MRI, were successfully able to monitor fibrotic changes in bleomycin-induced lung injury. The translational approach of using non-invasive imaging techniques show potential also from a clinical perspective.

## 1. Introduction

Fibrotic lung disease is a major health care burden and contributes to enormous morbidity and mortality globally [1,2,3]. In many cases, no causative agent can be identified, and idiopathic pulmonary fibrosis (IPF) is the largest group of patients where progressive lung fibrosis is the major pathological endpoint [4,5]. In addition, an increasing incidence of drug-induced interstitial lung disease (DIILD) is observed [6,7], and many different insults and chronic exposures to harming agents have been described to contribute to the development of fibrotic disease. Other causes for development of lung fibrosis can be due to familiar genetic disposition or simply environmental factors such as work-related exposure or smoking [1,8]. The common scenario for most cases is the presence of an initial phase of injury or acute inflammation. The repair process is mainly orchestrated by immune cells, initiating a wound repair mechanism which will restore the lung architecture to normal conditions. However, in some cases, the balance shifts from normal wound healing towards pathological fibrotic formation and symptoms like shortness of breath and chest pain start to occur. Severe lung fibrosis, if left untreated, often leads to death of the patient. Therefore, early diagnosis is very important as patients can be prescribed anti-fibrotic treatment in order to slow down disease progression before the loss of lung function, and the disease becomes too severe.

On a molecular level, fibrosis can be considered as scar tissue formation, a state where the tissue repair process has resulted in an excessive amount of extracellular matrix (ECM) deposition. In order to repair a wound or resolve the initial injury or the acute inflammatory process, certain markers, i.e., cells and molecules are known to be present [8,9,10]. Such key markers, known to play an important role during the initiation of pro-fibrotic processes, are e.g., TGF-β family members and the integrins [11]. Later on, the ECM proteins such as collagens are the key components of scar formation, and the enzyme lysyl oxidase (LOX) plays an important role in cross-linking the collagen fibers, making the scar robust and dense [12,13]. Most often, non-invasive monitoring or diagnosis of lung disease is done by imaging. Normally, X-ray scans and computed tomography (CT) are employed. These techniques are valuable in terms of reducing the need for invasive readouts such as bronchoscopy or biopsy sample collection [14,15]. However, progression of fibrotic lesions may be difficult to detect in CT-images at an early stage of the disease. Therefore, more sensitive and specific imaging biomarkers (IBs) are warranted.

Recently, an attempt to develop such specific IBs was presented using the bleomycin animal model of lung inflammation and fibrosis, where magnetic resonance imaging (MRI) was partially able to distinguish inflammation from fibrosis [16]. Even though MRI has the potential for promising IBs, that can be developed further, it lacks a unique signal for specific pathologies such as fibrosis. To address the specificity, a novel positron emission tomography (PET)-tracer for detection of newly synthesized and non-cross-linked Collagen-I, was developed and evaluated by the lab of Prof. Caravan and colleagues [17,18]. The tracer is a small peptide that binds to collagen fibrils. However, the specific binding site is only available before the collagen fibrils have been cross-linked into larger fibers. Hence, the primary peptide-binding site, available for the PET-tracer to bind, is not exposed in mature scar tissue, but only in immature scar active forming sites. Most likely, this active process is not detectable by CT or MRI. The collagen tracer has previously been employed in mouse studies [17,18], and also explored for the first time in a human study [19].

Other imaging methods such as MRI have been tested to assess fibrosis in animal models [20,21,22,23,24,25,26,27,28,29]. The bleomycin model, when instilled intratracheally (i.t.), is a common model for lung fibrosis. Investigations in the bleomycin model however are often focused on early time points (day 14–21), when the fibrosis is still accompanied by substantial inflammation. Also, fibrotic associated events can be visible later on, as shown previously by a multi-modality approach study where established fibrosis was mainly evident at day 28 [16]. Imaging in an early stage of the bleomycin model will involve a mixture of lesions consisting of inflammation, edema as well as simultaneous ECM build-up. Even in studies where the imaging is performed during the late stage of the bleomycin-induced pathology, there is a large intrinsic variability and a variation created by resolution or progression of the disease [16].

The aim of this study was to investigate how the combination of MRI and PET/CT imaging techniques can be used to monitor the progression of fibrotic disease in a rat model of lung fibrosis induced by bleomycin. More specifically, a PET-tracer specific for Collagen-I was evaluated for its ability to detect early formation of fibrotic lesions. Tissue histology and gene expression patterns were assessed, and analyses for collagen content and cross-linking were performed in an attempt to link the molecular mechanisms of lung remodeling with the PET-imaging results.

## 2. Experimental Section

### 2.1. Animals and Ethical Permit

Sprague-Dawley male rats (n = 58) were purchased from Janvier Labs (Le Genest-Saint-Isle, France). The rats were nine weeks old at the start of the experiment and were housed at Lund University or Medicon Village animal facilities with 12 h light/dark cycles and fed *ad libitum*. All animal studies were ethically reviewed and carried out in accordance with European Directive 2010/63/EEC and the ARRIVE guidelines [30]. The studies were approved by the local ethical committee in Lund/Malmö, Sweden, before they were initiated (ethical permit number 4003/2017 and 3226/2017).

### 2.2. Experimental Procedure

The rats were allowed to acclimatize to the housing conditions for at least 5 days, and a general health check was performed before start of the experiments. The rats were divided into two groups, (group I) the imaging group (n = 18) and (Group II) the non-imaging group (n = 40) (Figure 1). For the animals in Group I, a baseline scan by MRI was obtained. After bleomycin challenge, the animals were longitudinally imaged using a multi-imaging protocol (MRI and PET/CT) on days 7, 14, 21 and day 28. The animals in Group I were terminated on day 36 post-challenge. One rat per time point was removed from the longitudinal scan workflow and terminated to enable autoradiography of the lung. Rats (n = 4–6) from Group II were terminated at the same time points as in Group I followed by collection of lung tissue. Saline challenged rats were present in both groups as healthy controls. Additional information of the methodology is described in detail in the Appendix A.

### 2.3. Bleomycin Challenge

One single i.t. dose of bleomycin (Sigma Aldrich, St. Louis, MO, USA), was administered on day 0, with a concentration of 1000 iU, dissolved in 200 µL saline. Control animals received the same volume of saline. The procedure of i.t. administration was performed as previously described [16].

### 2.4. In Vivo Imaging

Animals were initially anesthetized using isoflurane at 4%, and during imaging isoflurane was maintained at 1.5–2%, delivered via a nose cone, in a 1:1 mixture of O_2_ and N_2_O. Longitudinal imaging was performed in Group I with PET/CT and MRI at Lund University BioImaging Centre (LBIC). Each rat was imaged by all modalities in one workflow before allowed to wake up after the scan session, which was repeated once per week, over four weeks. After PET-tracer injection, the MRI examination was performed, followed by the PET/CT examination. The PET imaging was initiated 1 h (+/−5 min) after the tracer injection. Imaging was performed non-gated. The procedure of multi-modality imaging has been previously described [16]. Further details are given in Appendix A.

#### 2.4.1. MRI

Rats were imaged on a preclinical 9.4T MRI Biospec AV III using the software ParaVision 6.0.1 (Bruker, Ettlingen, Germany). The MRI protocol has been described previously [16]. Briefly radial Ultra Short Echo time (UTE) sequences were used with two different echo times (TE); short (TE_SHORT_) with 0.324 ms and long (TE_LONG_) with 1 ms. The TE_SHORT_ sequence reflects the total MRI signal in the lung, derived from vessels and lesions with both inflammatory (edema) and fibrotic components. The TE_LONG_ sequence mainly reflects the fluid signal from vessels and inflammatory lesions. The field of view (FOV) was 58 × 58 mm^2^, with a matrix size of 192 × 192 and the repetition time (TR) was 8 ms. Further detailed description of scan parameters is summarized in Appendix A.

#### 2.4.2. PET/CT

The radiochemistry was performed at the LBIC laboratory by conjugation of the small collagen peptide referred to as CBP (Collagen Medical LTD, Boston, MA, USA) using the adjuvant NODOGA, with the radionuclide Cu^64^, produced at the Denmark Technical University (DTU) laboratory at Risø, Denmark. The radiochemistry protocol was employed according to the work by Désogère et al., [17,18] with minor in-house adjustments, as described in detail in Appendix A. After the labeling process, purification, and quality validation, the tracer CBP-Cu^64^ was delivered to the imaging laboratory with a total activity of approximately 800 MBq, and subsequently diluted in Saline 0.9%. Each syringe was prepared from the diluted stock just before injection. CBP-Cu^64^ was injected intravenously via the rat tail vein at a dose of 35 (±5) MBq in a total volume of 200 µL saline. Scan protocols and dose were tested initially during a pilot study. Also, acquisition time post-injection was optimized before the main study. Organ distribution of the tracer was acquired during in vivo PET imaging and, in addition, organ distribution of the tracer was measured ex vivo, where organs were collected in tubes and radioactivity was measured as counts on an Automatic Gamma counter (WIZARD^®^ 1480 RiaCalc WIZ, PerkinElmer, Life and Analytical Sciences, Shelton, CT, USA). The pilot study data and images are presented in Appendix A.

During the main study, MRI examination was performed before the PET/CT, starting immediately after PET tracer injection. Directly after the MRI, the animal was transferred and connected to the PET/CT system; (nanoScan^®^ PET/CT, Mediso, Hungary) for CT and PET imaging. Initially, a first overview scan (scout-view) with 11 s acquisition was performed to confirm optimal position of the animal. Once the optimal FOV was obtained covering the lungs, a CT scan was performed at a total scan time of 9 min, with a voxel size of 0.14 × 0.14 × 0.14 mm^3^, after reconstruction. PET imaging was finally performed, 1 h (+/−5) min post-injection. The PET scan acquisition time was 20 min, thereafter the animal was allowed to wake up from anesthesia. Post-reconstruction of the PET data was performed using a voxel size of 0.4 × 0.4 × 0.4 mm^3^ and CT-images were used for attenuation correction and anatomical registration. Additional CT and PET scan parameters are summarized in Appendix A.

### 2.5. Imaging Data Analysis

All acquired images were analyzed using the software VivoQuantTM 3.5 (inviCRO Imaging Services and Software VivoQuant, Boston, MA, USA). Initially, the acquired images were qualitatively evaluated and thereafter image analysis was initiated by drawing the regions of interest (ROI) semi manually for the lung volume in each MRI slice (Lung-ROI). The lesions were extracted according to a method described previously [16]. Briefly, the lesion ROIs in the MR-images for both echo-times respectively, were defined as the high signal area within the Lung-ROI, defined by histogram analysis. The vessel signal was subtracted in both scans and the high signal from TE_LONG_ mainly reflects the fluid present in the inflammation areas, and hereafter referred to as edema. The “tissue” signal indicating fibrotic tissue was assessed by the signal remaining from the subtractions of the TE_SHORT_ and TE_LONG_ ROIs. The ROIs from the MRI were subsequently transferred to the PET/CT frame of reference. In the PET images, the signal uptake was measured in the total Lung-ROI as well as the same ROI as was obtained for the lesions in the MR-images. This enabled the visualization of PET signal uptake in the mainly inflammatory lung regions (by TE_LONG_) and the PET signal uptake in the areas including both inflammation and elements of fibrotic streaks (TE_SHORT_). Data extraction was thereafter performed from these different modalities using the ROIs. A thorough detailed description of image data processing and Lung-ROI definitions can be found in the Appendix A.

### 2.6. Termination of Experiment and Sample Collection

The animals from Group I were terminated one week after the final imaging session was completed (day 35). Rats from the non-imaging group were terminated sequentially during the experiment, at day 3, 7, 14, 21, 28, 35, and 42 post-bleomycin administration (Group II). At termination, an intraperitoneal overdose of Pentobarbital Sodium (Apotek Produktion & Laboratorier AB, Stockholm, Sweden) was injected and lungs were harvested. The left lung was insufflated with a 4% paraformaldehyde fixative solution via the trachea while the right side of the lung was ligated. Thereafter, the remaining lobes were dissected, weighed, and snap frozen in −80 °C until use.

### 2.7. Hydroxyproline Analysis

Hydroxyproline content of the right middle lung lobe was assessed using the Hydroxyproline Colorimetric Assay Kit (Bio Vision, K555) according to manufacturer’s protocol. In short, lung tissue was homogenized in a Bead Mill 24 (Thermo Fisher Scientific, Hampton, New Hampshire, USA), using seven 2.8 mm ceramic spheres per vial, and hydrolyzed in 6N hydrochloric acid for 3 h at 120 °C. Thereafter, 5 µL per hydrolyzed sample was transferred to a clear flat bottomed 96 well plate and left to evaporate to dryness at 60 °C. Samples and standards were submitted to a reaction with chloramine T and DMAB reagent and absorbance was measured at 560 nm using SpectraMax i3x (Molecular Devices).

### 2.8. Lysyl Oxidase (LOX) Activity Assay

LOX activity was measured in the right inferior lung lobe using the fluorometric Lysyl Oxidase Activity Assay Kit (Abcam, ab112139) according to manufacturer’s protocol. In brief, 1 mg/mL total protein from homogenized lung tissue (see above for hydroxyproline analysis) was incubated with LOX reaction mixture for 30 min at 37 °C in a black opaque flat bottomed 96 well plate. Recombinant human LOX homolog 3 protein (R&D systems, 6069-AO) was used as a standard. Fluorescence was measured at excitation and emission wavelengths 540 and 590 nm respectively using a SpectraMax i3x (Molecular Devices).

### 2.9. Autoradiography

Autoradiography was performed for assessment of distribution of radioactivity within the lung. One rat at each time point was sacrificed directly after the imaging session and lungs were dissected for autoradiography analysis. Different lung lobes were placed in a mounting-mold and covered with embedding medium (TissueTek-OCT). The mold was then placed directly upon dry ice, freezing the lungs with embedding medium. The frozen samples were immediately taken to the cryotome and sectioned. The sample thickness was adjusted to 20 µm, to obtain sufficient signal from the radionuclide activity to the autoradiography instrument. Representative sections were collected, and adjacent sections of 10 µm thickness saved for histology staining with Hematoxylin and Eosin (H&E). Subsequently, the slide was placed inside the measurement device, a silicon-strip detector-based instrument with a 50 µm intrinsic resolution (Biomolex 700 Imager) to assess radioactivity distribution in the sections, as previously described [31]. Imaging time was prolonged until satisfactory image quality was reached (18–48 h).

### 2.10. Histology Staining and Analysis

After fixation and paraffin embedding, the left lung lobe was sectioned into 4 µm thick sections. The whole left lung was sectioned in the sagittal plane at four positions, to obtain good visualization of the widespread lesions throughout the whole lobe, section I–IV. The sections were then stained with H&E and Sirius Red staining, using a staining kit (Picro Sirius Red Stain Kit, ab150681, Abcam). All sections were evaluated by two observers and scored independently and blindly using modified Ashcroft score [32]. Total scores were presented graphically, and representative images were presented for each evaluated time point (day 28, 35, and 42).

The paraffin-embedded and sectioned tissue slices were also immunohistochemically stained for Collagen-I. The tissue sections were first cleared and rehydrated, followed by antigen retrieval. Then, peroxidase blocking was performed using a H_2_O_2_ and methanol mixture in Tris-based buffer solution. After washing, the sections were further blocked with 2% serum, following overnight incubation at 4 °C with the primary antibody directed towards Collagen-I (Thermo Fisher Scientific, Waltham, MA, USA), made in rabbit, at a dilution of 1:200. After rinsing the sections and incubating with a biotinylated secondary anti-rabbit IgG antibody supplied within the Vectastain kit (Vectastain^®^ Elite^®^ ABC kit, Vector Laboratories, Peterborough, UK), the staining was visualized with 3,3′-diaminobenzidine (DAB) supplied within the kit (ImmPACT^®^ DAB, Vectastain, Vector Laboratories, Peterborough, UK) and counterstained with methyl green (Methyl Green Counterstain, H-3402, Vector Laboratories, Peterborough, UK). Finally, the stained sections were dehydrated and mounted using Pertex (Histolab Products AB, Askim, Sweden).

### 2.11. Gene Expression Analysis from Lung Tissue Homogenates

The frozen postcaval lobe was homogenized and gene expression was analyzed. To obtain approximately 30–40 mg lung tissue homogenates, the gentleMACS™ Dissociator (Miltenyi Biotec, Auburn, CA, USA) was used with addition of β-mercaptoethanol (Sigma) in lysis buffer supplied within the RNA extraction kit (RNeasy Mini kit, Qiagen). The RNA was extracted and the concentration was measured by nanophotometer p330 (IMPLEN, INC. CA, USA), then 2 μg of total RNA, which equal to 50 ng cDNA per reaction, was reverse transcribed into cDNA using the reverse transcription-kit RT2 from Qiagen. Thereafter, the genes of interest were analyzed by mixing cDNA, Mastermix (Qiagen), and corresponding forward and reverse primers (Sigma Aldrich) and by using standard thermocycling on a Bio-Rad system (CFX96TM Real-Time System, C1000TM Thermal Cycler), PCR products were assessed. Data processing was performed using the CFX Manager^TM^ Software (Bio-Rad Laboratories, Inc, Hercules, CA, USA) and exported to Excel (Microsoft Office Professional Plus 2016) for further analysis. The forward and reverse genetic sequences are presented in Appendix A. The following genes of interest were analyzed: TGF-β I, TGF-β II, TGF-β III, integrin-β6, gremlin1, α-SMA, collagen IαII, collagen IIIαI, LOX, TIMP1, serpine1, serpin H1, IL-1β, Muc1, and periostin. For quantification of gene expression, the ΔΔCT-method was applied [33]. Genes of interest were calculated in relation to the geometric mean of two reference genes; β-2-macroglobulin (β2M) and Receptor-like protein 13 (RLP13a). Gene expression was then normalized to its control sample at each time point. Data was presented as mean ± standard error of the mean (SEM).

### 2.12. Statistical Analysis

Data are expressed as mean values and SEM unless otherwise specified. All data was analyzed using non-parametric tests using the software GraphPad Prism (version 8.4.3 GraphPad Software, San Diego, CA, USA, (www.graphpad.com) and IBM SPSS (version 23, IBM, Somers, NY, USA). To analyze differences between groups, One-way ANOVA test was applied with following post-hoc Bonferroni’s multiple comparisons test. Mann–Whitney test was used to compare variance between groups. *P*-values of less than 0.05 were considered statistically significant. Significance was indicated by * when *p* < 0.05; *p* < 0.01 by **; *p* < 0.001 by *** and *p* < 0.0001 by ****, when comparing bleomycin towards the saline control from the same time point. The comparison of different time points between bleomycin-challenged groups were expressed as # when *p* < 0.05; ## when *p* < 0.01; ### when *p* < 0.001 and #### when *p* < 0.0001. For comparison to baseline, within the same group of animals, significance was indicated by § when *p* < 0.05; *p* < 0.01 by §§; *p* < 0.001 by §§§ and *p* < 0.0001 by §§§§, as in the case of longitudinal measures, e.g., total lung volume or MRI signal.

## 3. Results

### 3.1. Model Characterization and Health Check Parameters

The bodyweight decreased during the first week for all animals exposed to bleomycin, but increased for the remaining time of the experiment, and reached a similar bodyweight as the saline control animals at the end of the experiment (Figure 2A). The total lung volume increased in bleomycin-challenged animals, mainly during the first week, compared to controls, and continued to increase at a slow pace throughout the experiment (Figure 2B). This pathological lung volume increase is also evident when presenting the lung volume to bodyweight ratio, where bleomycin challenged animals showed a substantial increase compared to controls already during the first week (Appendix A). The lung weight assessed at each termination point increased initially and remained at a higher level for the bleomycin-exposed animals compared to controls (Figure 2C). Evaluation by modified Ashcroft score indicate fibrosis up until 42 days post-challenge (Figure 2D). The stained sections (I–IV) evaluated at each time point indicate a heterogeneous and patchy disease. The scoring is expressed as minimum to max values of each section and per evaluated time point, expressing rather low mean score yet indicating high scores in selected areas within the tissue section evaluated. Representative images from H&E- (Figure 2E) and Sirius Red (Figure 2F) stained lung sections showed evident fibrotic regions alongside less affected areas, involving minor alveolar wall thickening, for time points 28, 35, and 42 days post-bleomycin challenge.

### 3.2. Lesion Assessment by CT and MRI

CT-images acquired of the thorax at different time points during the multi-modality imaging, indicated substantial lesions appearing as fuzzy grey areas within the lungs, and clearly visible compared to the controls, as shown in Appendix A.

The axial MR-images of the thorax display the lungs as dark areas. The muscle tissue, heart, liver, vessels, and lesions are depicted as bright objects (Figure 3A). The lung volume originally segmented for data extraction (Figure 3B), was assessed by histogram analysis. The extracted lesion volume from the TE_SHORT_ and TE_LONG_ MR-images resulted in slightly different areas (Figure 3C). MRI measures expressed a distinct edema peak at day 7 (Figure 3D) and a tissue peak at day 28, indicating mostly acute inflammation during the first week of this model, while fibrosis appears later (Figure 3E).

### 3.3. Using the PET-tracer CBP-Cu^64^ to Assess Non-Cross-linked Collagen-I

Uptake of the novel PET-tracer CBP-Cu^64^ in lung tissue was assessed at day 7, 14, 21, and 28 post-bleomycin or saline (as control) i.t. administration. Representative PET images of the CBP-Cu^64^ show uptake in the lungs at all measured time points post-challenge (Figure 4A). There was a higher total uptake of the tracer in the lungs of bleomycin-exposed animals compared to controls (Figure 4B). Large variation of the uptake within the bleomycin group was observed, especially at later time points. From a detailed visual inspection, it was evident that the major PET signal did not overlap directly with the MRI-identified lesions. Specifically, the PET tracer uptake was often localized at the border of the lesions identified by MRI. The PET signal evaluation from the various compartments (lesion regions vs. surrounding healthy appearing tissue), indicated a slowly increasing area of active fibrosis (Figure 4C,D). This observation was further confirmed in terms of tracer uptake in the tissue on a microscopic level by autoradiography, showing mainly tracer uptake occurring in dense areas (i.e., identified as tissue signal by MRI) but also in non-fibrotic areas close to fibrotic foci, as confirmed by H&E stained sections (Figure 4E).

In addition, correlation plots were achieved comparing PET uptake and MRI signal. Even though, the MRI-signal and the PET uptake of the tracer did not spatially overlap completely, as presented from total- and regional signal uptake, there was a positive correlation presented within the total lesion ROI, Appendix A.

### 3.4. Molecular Assessment of Collagen Production and Cross-Linking

The total collagen content assessed by the hydroxyproline assay increased over time, compared within the bleomycin challenged group, and it was also significantly greater than control at day 14 and onward (Figure 5A). In addition to the total collagen content, the collagen cross-linking ability was assessed by measuring the enzyme activity of LOX, known to be involved in cross-linking of collagen. The LOX levels also increased over time, during the four weeks’ model (Figure 5B), although not statistically significant. Lung tissue sections stained for Collagen-I indicated higher levels of collagen in the bleomycin challenged rats compared to controls, at all investigated time points (Figure 5C). Overall assessment of the stained sections suggests increased collagen content mainly around the larger airways at day 28. At day 35, the positive collagen stained areas included more regions with small airways and also being more evenly distributed throughout the parenchyma. At day 42, denser areas of positive collagen staining were seen, while also large unaffected regions could be observed.

### 3.5. Observed Gene Expression Levels in Bleomycin Challenged Rats from Inflammation to Fibrosis

Several pro-fibrotic markers showed a trend towards upregulated levels in the bleomycin group compared to control. The findings were not statistically significant, most probably due to the small sample size n = 4–5. All three TGF-β variants were induced mainly at day 14 and onward (Figure 6A–C). Integrin β6 was progressively upregulated over time (Figure 6D), peaking on day 28, while gremlin1 (Figure 6E) and α-SMA (Figure 6F) were mainly upregulated at day 14. Several other fibrosis-associated genes were induced at day 14 e.g., Collagen-I and III, LOX, periostin, serpine1, and serpineH1 (Figure 6G–L). IL-1β and TIMP1 were induced during the first three days (Figure 6M,N), while muc-1 was significantly upregulated over time and peaking at day 28 (Figure 6O).

## 4. Discussion

The aim of this work was to investigate how the combination of PET/CT and MRI imaging techniques can be used to monitor the progression of fibrotic disease in a rat model of lung injury induced by bleomycin. The fibrotic processes in this model were assessed by using a multi-modality approach including a novel PET-tracer developed for Collagen-I recognition together with multi-echo MRI identifying inflammatory and fibrotic lesions. In addition to IBs of fibrotic disease, also gene expression profiling and histological evaluation were performed at multiple time points in order to confirm fibrosis establishment. The initial characterization of the model showed distinct bodyweight loss of bleomycin challenged rats and a marked increase in total lung volume during the first weeks, evident from terminal lung harvest endpoints as well as during longitudinal MRI lung volume measurements. The increase in total lung volume has previously been shown, particularly in the bleomycin i.t. challenge model [16,25,26,29,34,35], but it has also been reported in other types of lung injury models [34,35,36,37]. This phenomenon is thought to be a compensatory mechanism to cope with the acutely induced vascular leak [35,38]. In the bleomycin model, this is mainly occurring during the first week after one i.t. dose of bleomycin, which results in lesions of edema and enormous cytokine release, cellular infiltration, and activation, alongside collagenases and proteases being released, causing further damage locally [16,20,39,40].

In a clinical setting, CT is the most employed imaging modality for assessing lesions in ILD and fibrosis [41,42]. As CT is mainly available and cost-effective, most diagnostic and prognostic scans are repeatedly done in patients in this modality. However, from the CT-images, which mainly reflect the tissue density (electron density), it is difficult to distinguish between inflammation and fibrosis. In animal models of lung diseases, MRI is often preferred to CT due to its many contrast mechanisms. In this study, we have explored different imaging modalities, PET/CT and MRI, and combined them in order to assess inflammatory and fibrotic lesions in the bleomycin-induced lung injury model in rats. The MRI scans in this study were employed according to previously published work [16], which indicated a rather clear time course for when the inflammation peaked and when the fibrotic lesions dominated, i.e., in day 7 and 28, respectively.

From the results of the PET measurements, it was evident that the bleomycin-challenged rats expressed a larger fraction of tracer uptake in relation to the injected dose, compared to healthy controls. However, there the accumulated level of CBP-Cu^64^ was similar over the time period studied. This tracer is able to detect Collagen-I, while not being cross-linked nor intertwined in larger collagen fibers [17,18]. To further explore the analysis, the uptake in different compartments of the lung was performed, over time. For this evaluation, we used MRI to define the lesions (using histogram-based thresholding) and assessed the PET-signal within the inflammatory lesions, fibrotic lesions, and healthy areas around the detected lesions. The PET-signal seemed to be detected in close vicinity but mainly outside the MRI-defined lesions. This pattern increased for later time points post-challenge, indicating dynamic pro-fibrotic processes mainly in the borderline of the MRI detected lesions. This would be the active sites of newly synthesized and assembled collagens fibrils, and the area where the Collagen tracer CBP-Cu^64^ is mainly accumulating. This is in line with the findings from the human study published by Montesi et al., [19], exploring the CBP-tracer. In this study, it was evident that the tracer could be detected in lesions also detected by CT, as well as adjacent areas of seemingly healthy lung tissue. This indicates early detection of active fibrotic processes by the PET tracer, although not yet visible by CT imaging. In our study, in addition to the in vivo MRI and PET/CT imaging, the PET tracer uptake was assessed by autoradiography, in lung sections. A similar observation was made as from the MRI and PET/CT approach, i.e., the PET signal was mainly located in the borderline of dense fibrotic lesions next to tissue areas with less damage and alteration, and not only within fibrotic lesions.

The total uptake of CBP-Cu^64^ tracer in the lungs of bleomycin challenged rats, seemed to increase slowly with time (mean uptake at day 14–28) while also large individual differences of the total activity were found. However, already at day 7, the uptake of the collagen tracer was considerably high. The levels of gene expression of Collagen-I peaked at day 14, which would point towards newly synthesized collagen occurring later than day 7. The existing collagen within the lung is normally going through turnover processes, with a continuous release of enzymes able to break down collagens and induced cells that could subsequently clear that debris [43,44]. As the acute inflammatory phase occurs during the first week of the bleomycin model, one could speculate that this leads to excessive release and activation of catalytic enzymes such as collagenases, peptidases, and Matrix metallopeptidases (MMPs) to degrade the existing collagens holding up the lung tissue structure [45,46], and thus expose the CBP peptide binding site. Accordingly, speculating that the CBP-attractive sites become available for peptide attachment, during destruction of the collagen due to the huge release of proteins and collagenases in the phase of acute injury, tissue damage and vascular leak occurring initially from local bleomycin challenge. This might explain the reason for the large, yet clustered presence of tracer uptake already at day 7 [43,44,45,46]. As pro-fibrotic processes are known to be initiated during the second week and further on, including new collagen synthesis, the ECM excess is building up with increased collagen along with tissue fibrosis. This was confirmed by the increasingly high collagen production over time, as indicated by hydroxyproline assessment. In addition, we also investigated the cross-linking activity of the LOX in our model. These analyses point towards continuous LOX activity, contributing to the active collagen cross-linking, however indicating large variability within the bleomycin group at day 28. The large variability of hydroxyproline content and LOX activity is in line with high variability of PET tracer uptake, indicating active fibrosis in some individuals, whereas others seem to be resolving the injury.

Additionally, the gene expression analysis confirmed the pro-fibrotic markers being actively increased during the later time points. Some of the most important ones being the TGF-β family members as well as the integrins, known to play crucial roles in fibrosis establishment [11]. Also, clinical studies have revealed important novel markers associated with fibrotic disease such as an increase of periostin, Muc1 and Gremlin [47,48,49]. Similarly, we have observed a marked increase of these genes in the present study, which are mainly upregulated in our model after day 14 and onward. However, Serpine1 (PAI-1) and TIMP1 genes were seen to be upregulated already during the first week of this model. Both these markers are linked to the ECM degradation process and known to be involved in the wound healing process after tissue insult, following fibrosis [50,51].

Tissue staining by H&E and Sirius red revealed progression of fibrotic areas with time. Interestingly enough, the location of the fibrosis seems to be shifting over time. At day 28, fibrotic (as observed by H&E and Sirius red staining) and collagen stained areas seemed mainly focused around larger airways, while later in the process, the fibrotic loci increased in size and were located more around the small airways and peripheral parenchyma. The sections collected at day 42 included dense fibrotic loci but also rather unharmed alveoli. This is unfortunately not displayed when only presenting the mean values from the scoring evaluation. Presenting the data from minimum to maximum values gave a better understanding of how the lesions were distributed and the establishment of the disease.

Limitations to consider in this study is the acute model as such, where locally administrated bleomycin induces toxic damage and more resembles a response to acute damage rather than the features of true fibrotic disease. To further improve this physiological lung injury model, it would be an advantage if bleomycin reached the lung tissue through the vasculature. This could be achieved by employing a systemic approach as in the case of bleomycin being repeatedly administrated via the intraperitoneal or subcutaneous route [52,53]. Therefore, a robust chronic exposure model of established fibrosis in animals should be considered more in the future. Not least, using other alternative fibrosis-inducing agents may be required to progress the field further. In addition, employing multi-imaging modality in these models could be a successful approach for development of non-invasive translational imaging biomarkers. In addition, the Collagen-I tracer would be interesting to use in combination with the PET tracer ^18^F-Fludeoxyglucose (FDG), as FDG has recently been acknowledged for its ability to detect both inflammatory areas and fibrotic lesions [54]. The FDG signal in fibrotic lesions in PET-studies have been attributed to the so-called Warburg effect recently identified in active fibrosis [55].

In summary, this work combines several non-invasive imaging modalities, PET/CT and MRI, for longitudinal assessment of bleomycin-induced lesions in the lungs over four weeks. The different examinations performed on the same animal using these modalities enabled unique data extraction from each modality by combining the information. In addition, terminal samples were collected from selected animals sequentially throughout the time course as the longitudinal imaging occurred, allowing for assessment of gene expression as well as collagen content and cross-linking, throughout the experiment.

## 5. Conclusions

In conclusion, the combined dual echo MRI and a novel PET tracer for imaging of Collagen-I demonstrates a successful approach to monitor early and late stage fibrotic changes in bleomycin-induced lung injury. By using non-invasive imaging techniques and scanning protocols for lung lesion assessment, our data suggest that IBs have the potential to further improve the understanding of ongoing pathological processes. Preclinical studies can thus support development of IBs and their further introduction into clinical use where they can be used to improve diagnoses and choice of therapeutic.

## Figures and Tables

**Figure 1 jcm-09-03706-f001:**
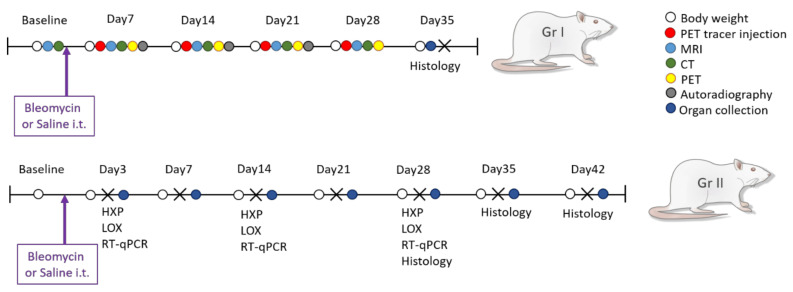
Study layout of bleomycin-induced lung injury and longitudinal imaging. Study layout with sampling and endpoints for the imaging group (Group I) and non-imaging group (Group II). Longitudinal imaging was done by combining MRI, CT, and PET in group I, while termination of 4–5 rats occurred every week, from Group II, for tissue sample collection and analysis of collagen content by hydroxyproline assay (HXP), lysyl oxidase (LOX) activity, gene expression analyses by RT-qPCR, and histological assessment by staining of lung tissue sections.

**Figure 2 jcm-09-03706-f002:**
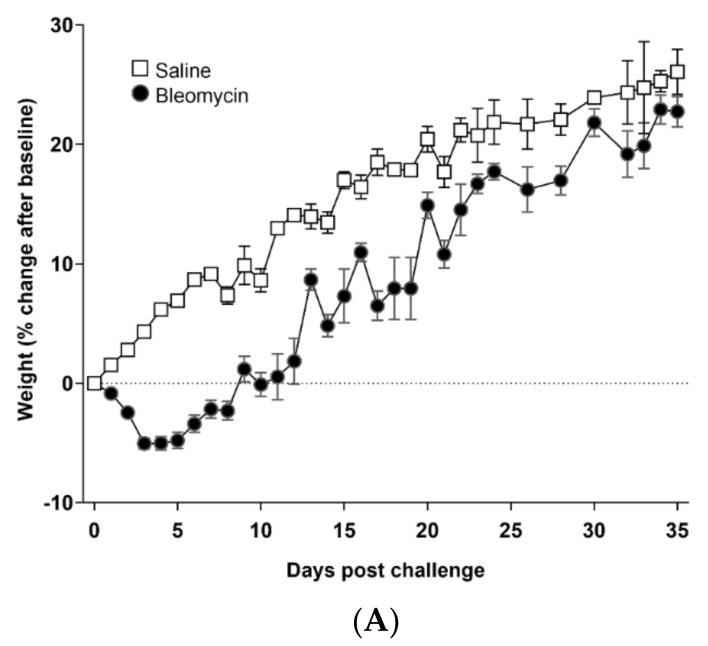
Model characterization and measures of basic parameters. (**A**) Bodyweight changes over time, (**B**) Total lung volume over time, assessed by MRI, and (**C**) right lung lobe weigh at termination (from Group II). (**D**) Fibrosis scoring evaluated by the modified Ashcroft score at day 28, 35, and 42 post-bleomycin challenge. Representative images of lung tissue sections stained by (**E**) H&E and (**F**) Sirius Red.

**Figure 3 jcm-09-03706-f003:**
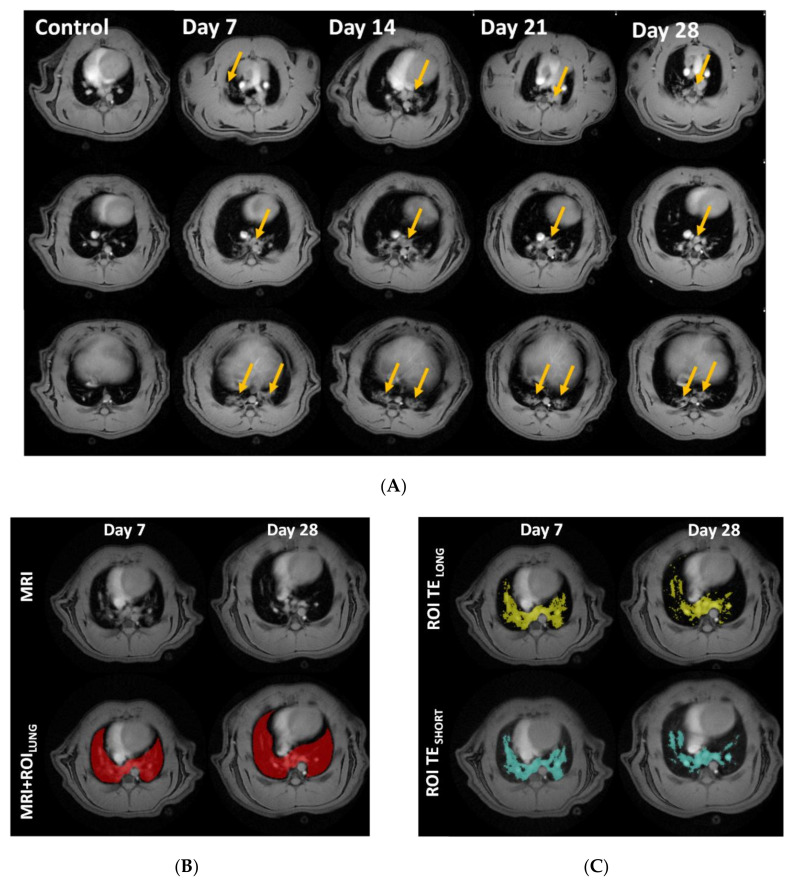
Longitudinal magnetic resonance imaging (MRI) in Group I, during 28 days of bleomycin-induced lung injury. (**A**) Representative MRI images of lungs from rats challenged by i.t. administration of bleomycin (or saline as controls), on day 7, 14, 21, and 28 post-challenge. Yellow arrows point out identified lesions (**B**) MRI defined Lung-ROIs and (**C**) UTE_SHORT_ and UTE_LONG_ defined regions in yellow and blue, respectively. Quantification of the lesion signal assessed by the two different MRI sequences was calculated for (**D**) “edema” signal indicating peak inflammation in bleomycin challenged rats at day 7, while (**E**) the calculated “tissue” signal appears to peak at day 28.

**Figure 4 jcm-09-03706-f004:**
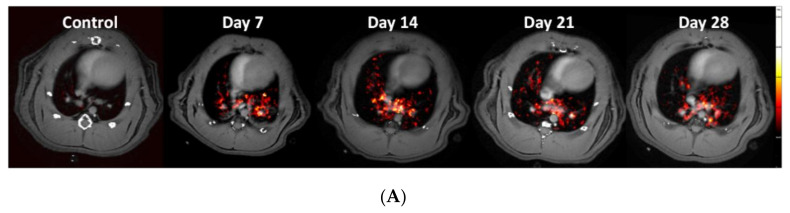
Longitudinal PET imaging in Group I, during 28 days of bleomycin-induced lung injury, using the novel PET tracer for assessment of non-cross-linked Collagen-I. (**A**) Representative PET images of lungs from rats challenged by i.t. administration of bleomycin (or saline as controls) on day 7, 14, 21, and 28 post-challenge. (**B**) Quantification of the total lung tissue signal uptake expressed as fraction uptake (total lung uptake of the injected dose [%ID]). The PET signal was assessed in different compartments, defined as high signal regions based on the TE_SHORT_ and TE_LONG_, sequences from obtained MRI scans. (**C**) PET signal uptake from TE_LONG_ defined lesions (yellow area) and surrounding area within the total Lung-ROI. (**D**) PET signal uptake within the TE_SHORT_ defined lesions (blue) and surrounding area within the total Lung-ROI. (**E**) Autoradiography was performed in few of the animals to assess microscopic tracer uptake directly after the PET scan. One time point (day 21) from the bleomycin-challenged group is presented here. Lungs R1–R4 are Right lung lobes from the top section and downwards. Left referring to the left lung lobe. Enhanced signal uptake of CBP-Cu^64^ is indicated by arrows (in the case of long streaks of signal) or encircled (in the case of focused area), where also fibrotic tissue is observed in close proximity to healthy tissue and vessel-airway presence.

**Figure 5 jcm-09-03706-f005:**
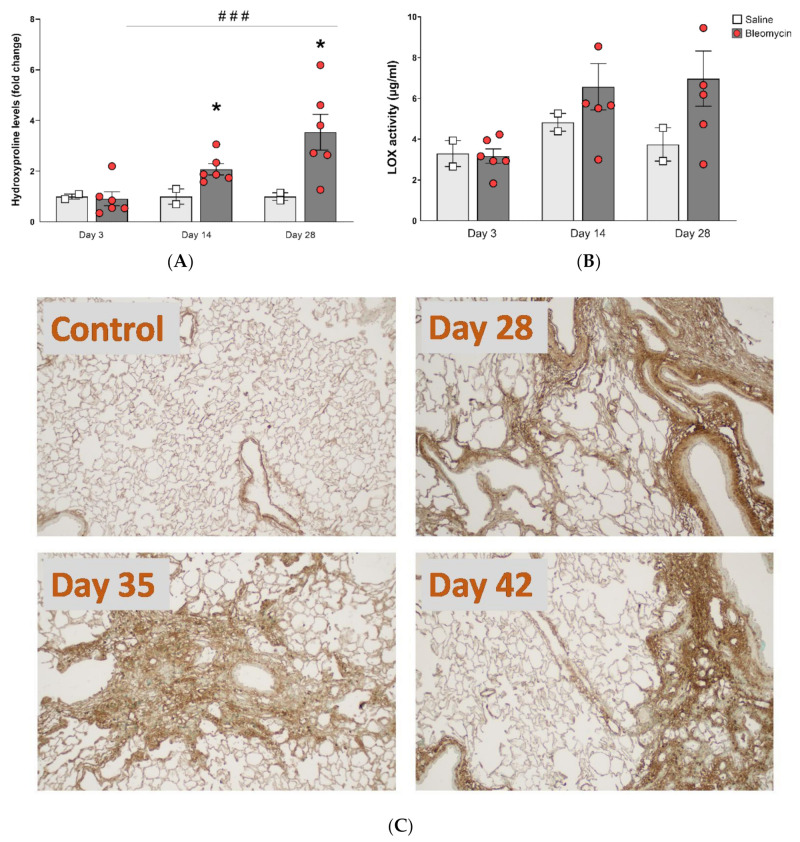
Collagen content and cross-linking activity. (**A**) Hydroxyproline content and (**B**) LOX activity, both increased in bleomycin challenged lungs compared to controls, and continued to increase over time. (**C**) Immunohistochemically stained tissue sections were acquired using Collagen-I antibody and representative images from three different time points are presented.

**Figure 6 jcm-09-03706-f006:**
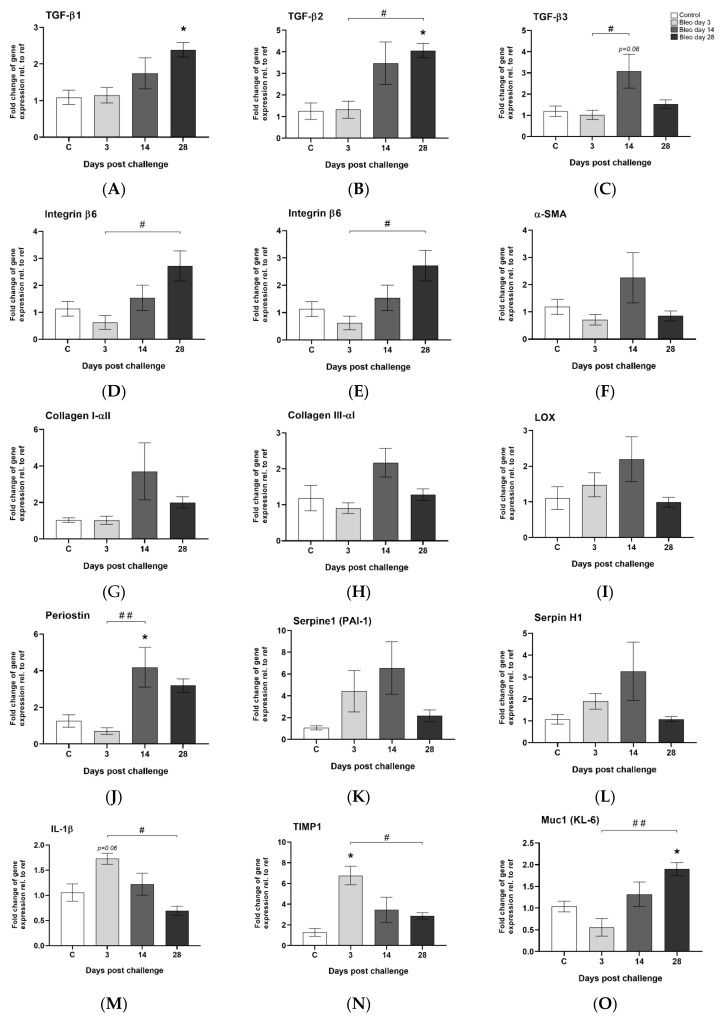
Gene expression of markers involved in lung injury and wound healing. (**A**) TGF-β I, (**B**) TGF-β II, and (**C**) TGF-β III were induced at later time points, similar as (**D**) integrin β6. The fibrosis-associated genes (**E**) gremlin1 and (**F**) α-SMA were both mainly induced at day 14. The same appeared for (**G**) collagen-I and (**H**) collagen-III, (**I**) LOX, (**J**) periostin, (**K**) serpine1, and (**L**) serpineH1. Inflammatory-associated (**M**) IL-1β was not expressed at late time points. (**N**) TIMP1 gene expression was induced early during the first days of the model, while (**O**) the muc1 gene was significantly upregulated with time and peaked at day 28. All data are presented as mean ± standard error of the mean (SEM). All samples from the bleomycin group are related to the mean of control and normalized to the geometric mean of two different reference genes. Each presented group/time point contains n = 4–5.

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
