# Peer review of "Longitudinal Imaging Using PET/CT with Collagen-I PET-Tracer and MRI for Assessment of Fibrotic and Inflammatory Lesions in a Rat Lung Injury Model"

_jcm, 2020, doi:10.3390/jcm9113706_

Round 1
Reviewer 1 Report
The manuscript was well developed and argued and the combined use of dual echo MRI and a novel PET tracer for imaging of Collagen-I 527 seems to be useful to in monitoring early and late stage of fibrotic changes in bleomycin-528 induced lung injury and these tecniques potentially further improve the use of the sperimental tracer in clinical practice in the future
Author Response
Reviewer 1:
Q or comment:
The manuscript was well developed and argued and the combined use of dual echo MRI and a novel PET tracer for imaging of Collagen-I seems to be useful to in monitoring early and late stage of fibrotic changes in bleomycin induced lung injury and these techniques potentially further improve the use of the experimental tracer in clinical practice in the future.
A: The authors are thankful for the evaluation of the manuscript and appreciate the comments by the reviewer. The tracer is currently in development for clinical use; thus we believe it is important to evaluate the application in other animal models besides mice, where it was tested previously.
Reviewer 2 Report
Why the rats control group is so disproportionately large compared to the research group?Author Response
Reviewer 2:
Q or comment:
Why the rats control group is so disproportionately large compared to the research group?
A: According to our previous experiments; to be able to confirm statistical differences a certain number of controls is required due to the experimental set-up. We divided the imaging sessions over two days in order to enable feasibility and better logistics. In one day we could only perform 8 animal scans, as each scan session takes 1.5-2 h to complete for each rat. For robustness we included both controls and bleomycin challenged animals in each scan session (2 controls + 6 bleomycin per day).
Reviewer 3 Report
The authors present sequential collagen-I microPET/CT and MRI findings of animal model with bleomycin induced acute lung injury. The comprehensive experiments are well planned and presented.
The study follows the changes over time by in vivo imaging modalities, histologic confirmation, gene expression markers and markers of inflammation. The study is essentially a careful surveillance of bleomycin induced acute lung injury in animal model.
The idea and findings are very interesting, offering a view to the patho-physiology of lung injury and inflammation.
I do have several comments that could be considered by the authors to improve the manuscript.
- The manuscript contains high volume of information. Consider shortening the manuscript. (Example: Do you need to explain the indication markers/symbols in the text in the Statistical Analysis section?)
- The authors performed meticulous, elegant experiments to observe the phenomenon of inflammation and fibrosis. The findings and results should conclude with the findings. I believe suggesting potential for clinical use is stretching the results too far, especially as you do not study the correlation between the image findings and clinical outcome/recovery. Only the natural course of the acute lung injury is studied to a time point of your choice (terminated at designated time). No response to treatment or time to natural death is noted.
- Please clarify how you defined "inflammation" (as you intended to explain active inflammatory response) since some pathologists explain fibrosis to be one stage of inflammation.
- Please add brief explanation of why the MRI signals were high or low for the different histologic features for those not familiar with the mechanism of different MR signal intensities.
- Unless I missed it, I saw no MIP image of the collagen-I PET MIP image in the manuscript. In the supplement images, lung uptake was not visually evident. Is this because your target to background ratio was low in the PET images? It may be worth mentioning in the discussion section, as low target to background ratio is a big impediment to clinical application of the tracer, especially considering the relatively high numbers FDG shows.
- Figure 5A is obscured at the left margin.
- The authors state in the conclusion that the imaging has potential for further improvement and clinical implications. I must disagree on these points. Please add short discussion for 'how IB could further improve' and what the clinical implications may be. (As I mention in point 2).
Author Response
Reviewers 3:
Q or comment: The authors present sequential collagen-I microPET/CT and MRI findings of animal model with bleomycin induced acute lung injury. The comprehensive experiments are well planned and presented.
The study follows the changes over time by in vivo imaging modalities, histologic confirmation, gene expression markers and markers of inflammation. The study is essentially a careful surveillance of bleomycin induced acute lung injury in animal model. The idea and findings are very interesting, offering a view to the patho-physiology of lung injury and inflammation.
A: The authors are thankful for the comprehensive evaluation of this work, and are happy to improve the manuscript further by answering reviewer’s questions and making changes to the text.
Q or comment: I do have several comments that could be considered by the authors to improve the manuscript.
Q1: The manuscript contains high volume of information. Consider shortening the manuscript. (Example: Do you need to explain the indication markers/symbols in the text in the Statistical Analysis section?)
A1: At the expense of clarity around the procedures (animals, injection and imaging etc.) we could shorten some sections, however as this tracer has never been evaluated in a rat model before, we wanted to describe it in a way for others to be able to reproduce the data. Substantial work involving many different methods and analyses were performed. The tracer optimization and pilot studies for PET have already been moved to the supplementary sections. Also, some parts of the imaging description, primer-sequences for the gene expression data etc., are also available in the supplements only in order to minimize the volume of information. Considering the statistical analysis section, the various symbol definitions are mentioned here in order to reduce text amount instead of writing the symbol definitions in every figure legend.
We could further reduce the amount of text, if this is still a concern.
Q2: The authors performed meticulous, elegant experiments to observe the phenomenon of inflammation and fibrosis. The findings and results should conclude with the findings. I believe suggesting potential for clinical use is stretching the results too far, especially as you do not study the correlation between the image findings and clinical outcome/recovery. Only the natural course of the acute lung injury is studied to a time point of your choice (terminated at designated time). No response to treatment or time to natural death is noted.
A2: Thank you for this comment. We have now changed the wording in the text (in the Conclusion section). The reviewer’s point of not having treatment groups or following the animals until natural death, is correct. Nevertheless, the reason for clinical implications suggested, is that we have gained new insight on how this tracer works on a microscopic level – as compared to the mouse studies done previously, before the first-in-human study. The larger rat lung, compared to mouse, enabled us to study whole lung accumulation of the tracer with greater resolution. Besides studying whole lung accumulation of the tracer we were also able to observe local lesion sites by ex vivo techniques. These results would not have been possible to obtain in either mouse (too small lungs) nor in human lungs for detailed tracer-uptake assessment (as biopsy is associated with a considerable risk for a patient and it was not performed in the first-in human study). Even if our study may not facilitate the clinical breakthrough, it gives additional knowledge on the functionality of this PET tracer, that is already being used in patients.
Q3: Please clarify how you defined "inflammation" (as you intended to explain active inflammatory response) since some pathologists explain fibrosis to be one stage of inflammation.
A3: The “inflammation” detected at early stage in this model was characterized by an acute inflammatory response involving vascular leak (plasma exudation), substantial immune cell infiltration into the lung tissue (in parenchyma found in histological analyses) and in the lumen of the lungs (observed both in histological sections and bronchoalveolar lavage samples from previous work). This acute inflammatory response to bleomycin-challenged rats, is dominated by neutrophils and later also eosinophils, and macrophages are increased and results in lung edema. On the other hand, the event during the later stage of this model, the remodeling, rather involves macrophages and fibroblasts, and is relatively a quiet and slow progressing event (also more discrete in terms of the health status of the animals and signal seen in e.g. MRI images). This late stage is not as acute even though involving immune cells (as you refer to being another stage of inflammation). We have clarified this by adding the word “acute” in several places throughout the manuscript (abstract, intro, results etc.) and have made sure to mention the fibrotic process as a very much active and live event (i.e. highlighting that fibrotic tissue is not just a scar and not just fibroblasts being present in those lesions). We hope that the reviewer acknowledges these improvements and that they make the message clearer.
Q4: Please add brief explanation of why the MRI signals were high or low for the different histologic features for those not familiar with the mechanism of different MR signal intensities.
A4: The MRI signal arise from the protons of the water molecules in the tissue. This signal depends on the water content and decay during the acquisition of the signal, i.e. the echo time. Different tissues have different decay times. Thereby, a MRI protocol can be designed to enhance the different tissue types, e.g. water/edema containing lesions or fibrotic tissue deposition. In this study two different echo times were selected to enhanced inflammation and fibrosis, respectively. The methodology section (2.4.1 MRI- section and also 2.5 Imaging data analysis-section) briefly describes the two selected echo times and their application, making this explanation available for the readers.
Q5: Unless I missed it, I saw no MIP image of the Collagen-I PET MIP image in the manuscript. In the supplement images, lung uptake was not visually evident. Is this because your target to background ratio was low in the PET images? It may be worth mentioning in the discussion section, as low target to background ratio is a big impediment to clinical application of the tracer, especially considering the relatively high numbers FDG shows.
A5: The tracer uptake is very high in the liver and also the kidneys, due to the metabolism of the tracer and excretion via urine. This uptake is always present; weather we look at a healthy animal or one that received bleomycin. The uptake in the lung however differs between the healthy vs. non-healthy animals. This small difference is significant even if total up-take is relatively small (when comparing other organs). The supplementary images of PET uptake from the pilot study are presented in regard to optimization of image acquisition time points. The lung-uptake is indeed lower compared to liver and kidneys, as is evident in many other studies involving various types of tracer kinetics. The high uptake in other organs has also been demonstrated previously by Desogere et al., 2017 (Figure 2E in their publication; Sci Transl Med 2017 Apr 5;9(384): eaaf4696, PMID: 28381537). Our data is in line with their results. We also give the relative increase for healthy vs. non-healthy lung, which was our main focus.
MIP images for the PET were not presented in the figures in the manuscript although representative PET-CT-MRI images from each time-point were however given in Figure 4A to show and guide the readers to where and to what extend the signal was detected. The relative uptake (%of injected dose) was presented in Figure 4B.
Q6: Figure 5A is obscured at the left margin.
A6: Thank you for this feedback. The image must have been shifted towards the right page margin for some reason in that particular figure, in the word file. We have now corrected this.
Q7: The authors state in the conclusion that the imaging has potential for further improvement and clinical implications. I must disagree on these points. Please add short discussion for 'how IB could further improve' and what the clinical implications may be. (As I mention in point 2).
A7: (Also see answer in Q2 above) The tracer could always be improved in specificity and affinity from a chemistry point of view, as any other tracer-agent. Also, this tracer is already being used in the clinical setting – and the more in vivo knowledge that can be acquired in animal models the possibilities to improve tracers for use in the clinic increases. The sentence about IBs for clinical implication has been rephrased in the conclusion section.
Reviewer 4 Report
This study combines PET/CT and MRI imaging technique to evaluate the progression of fibrotic disease in a rat model of lung injury induced by bleomycin. They evaluated the fibrotic process using a multi-modality approach with a novel PET tracer for collagen I in-vivo. It is interesting for the authors to use MRI based ROI/Masks for PET signal quantifications. As the author mentioned in the paper as well as the reference paper, the MRI short TE images reflects the total signal from vessels and inflammatory tissue, including edema and fibrosis. The long TE MRI images represents fluid signal from vessels and inflammatory tissue edema. The difference of signal (shortTE - longTE) indicates fibrotic tissue. The author uses shortTE and longTE masks but not the difference ROI mask for the PET signal quantification. And they reach the conclusion that PET tracer uptake localized on the border of the lesions identified by MRI. First, the imaging resolution of the MRI and PET are different, could image registration cause issue in the ROI? Second, is the PET imaging acquisition gated for breathing? Could breathing related artifact cause artifacts in the quantifications?
Since the autoradiography shows similar PET findings, it is not clear to putting MRI and PET together in the analysis. IF the authors try to show the dynamic fibrotic process, they could just show the temporal mapping of either the PET or the MRI delta-TE maps.
The lung injury model induced by bleomycin has long been established with the ex-vivo study. And the current PET with collagen-I tracer correlates well with the autoradiography exam, which suggests its usage for fibrotic evaluation. The PET uptake pattern appears heterogeneous, which is consistent with nature of fibrotic changes. Maybe this is enough for an animal model paper. If it has been published already, adding MRI imaging to the analysis does not provide additional value.
Author Response
Reviewers 4:
Q or comment:
This study combines PET/CT and MRI imaging technique to evaluate the progression of fibrotic disease in a rat model of lung injury induced by bleomycin. They evaluated the fibrotic process using a multi-modality approach with a novel PET tracer for collagen I in-vivo. It is interesting for the authors to use MRI based ROI/Masks for PET signal quantifications. As the author mentioned in the paper as well as the reference paper, the MRI short TE images reflects the total signal from vessels and inflammatory tissue, including edema and fibrosis. The long TE MRI images represents fluid signal from vessels and inflammatory tissue edema. The difference of signal (shortTE - longTE) indicates fibrotic tissue. The author uses shortTE and longTE masks but not the difference ROI mask for the PET signal quantification. And they reach the conclusion that PET tracer uptake localized on the border of the lesions identified by MRI.
A: This is a correct interpretation of the masks and ROI and how we generated data extraction. In order to clarify in what stage the pathological phase is, at various time points – the subtraction was done simply in the MRI data. However, in order to identify any potential lesions by PET we wanted to keep the whole lung as possible uptake-area, or else we might risk missing out uptake of tracer in areas not identified as lesions by MRI (short-TE or long-TE).
Q1: First, the imaging resolution of the MRI and PET are different, could image registration cause issue in the ROI?
A1: The MRI and PET are different systems with different spatial resolutions. However, using co-registration by the software, we were able to overlay MRI and PET images with the same scale in the assessment of the data. This process was validated by phantom studies. Further, this was evaluated qualitatively in every scan session and slice post imaging by multiple reviewers among the authors of this work (IMP, LEO, MS). Furthermore, the results of the autoradiography confirm on a microscopic level, that the strongest signal of tracer accumulation, is located at the borderline of the lesions.
Q2: Second, is the PET imaging acquisition gated for breathing?
A2: Neither of the imaging acquisitions were triggered for breathing. We have yet done our best to sample data as well and accurate as possible. It was decided not to use triggering due to the prolonged acquisition time overall involving 3 different imaging modalities and would extend beyond the current 1.5-2 h in total during each rat’s scan session, repeated each week. By limiting the field of view to only cover the lungs during the PET acquisition we could increase the data collection in this narrowed view without prolonging the acquisition time. The MRI sequences using UTE are not as sensitive to motion artefacts. We also used MRI-acquisition with radial sampling and several numbers of excitations (averages), to reduce the motion effects.
Q3: Could breathing related artifact cause artifacts in the quantifications?
A3: Thank you for the feedback and this question. Considering the movement of the lung but also the heart as the nearest organ, we have taken precautionary processes. We are aware of the potential improvement of the images using respiratory gating. We believe that our used methods described during acquisition (see Q2 above) is sufficient to reduce the breathing effects, although we are aware of the difficulties to remove the effect of potential artefacts completely. Furthermore, certain actions were taken during the postprocessing segmentation. Even though we generated sharp and clear images without respiratory gating during the acquisition, we have kept large margins towards surrounding tissue (heart, muscle, ribs, liver and diaphragm) during the segmentation process. Using these precautions, for drawing the ROI and working with the different masks, we could ensure that the final Lung-ROI did not include possible motion artefacts in close proximity to nearby tissue not part of the lung.
Q4: Since the autoradiography shows similar PET findings, it is not clear to putting MRI and PET together in the analysis. IF the authors try to show the dynamic fibrotic process, they could just show the temporal mapping of either the PET or the MRI delta-TE maps. The lung injury model induced by bleomycin has long been established with the ex-vivo study. And the current PET with collagen-I tracer correlates well with the autoradiography exam, which suggests its usage for fibrotic evaluation. The PET uptake pattern appears heterogeneous, which is consistent with nature of fibrotic changes. Maybe this is enough for an animal model paper. If it has been published already, adding MRI imaging to the analysis does not provide additional value.
A4: The PET alone indicates high signal up-take averaged over the total lung at all time-points post disease initiation. MRI enabled us to confirm the different disease phases of acute inflammatory stage mainly being present during week 1-2 (up to days 7-14) and the fibrosis that started appearing later on (day 21-28 and onwards). The inflammatory stage and fibrotic stage identified by MRI could then be used to put the PET data in context, since the images were acquired in the very same animals over time with PET-CT and MRI. Therefore, we could come to the conclusion that the Collagen tracer is actively accumulated both during the acute inflammation as well as during progressive fibrosis. Therefore, both modalities were used to overlay images from different scans that could highlight various pathological phases of the model. In addition, the autoradiography helps us to answer the question of where the tracer accumulation is found mainly, while MRI answers when (inflammation vs fibrosis) in the disease model that PET signal is present.
Round 2
Reviewer 4 Report
n/a